# Sludge Derived Carbon Modified Anode in Microbial Fuel Cell for Performance Improvement and Microbial Community Dynamics

**DOI:** 10.3390/membranes12020120

**Published:** 2022-01-20

**Authors:** Kaili Zhu, Yihu Xu, Xiao Yang, Wencai Fu, Wenhao Dang, Jinxia Yuan, Zhiwei Wang

**Affiliations:** Key Laboratory of Clean Pulp & Papermaking and Pollution Control of Guangxi, College of Light Industry and Food Engineering, Guangxi University, Nanning 530004, China; zhukaili26@163.com (K.Z.); xuyihuzwj@163.com (Y.X.); yxiao0112@163.com (X.Y.); fl392597@163.com (W.F.); wenhaodang14@126.com (W.D.); yuan09@syr.edu (J.Y.)

**Keywords:** sludge carbon, extracellular polymeric substance, microbial fuel cell, electroactive biofilm, microbial community dynamics

## Abstract

The conversion of activated sludge into high value-added materials, such as sludge carbon (SC), has attracted increasing attention because of its potential for various applications. In this study, the effect of SC carbonized at temperatures of 600, 800, 1000, and 1200 °C on the anode performance of microbial fuel cells and its mechanism are discussed. A pyrolysis temperature of 1000 °C for the loaded electrode (SC1000/CC) generated a maximum areal power density of 2.165 ± 0.021 W·m^−2^ and a current density of 5.985 ± 0.015 A·m^−2^, which is 3.017- and 2.992-fold that of the CC anode. The addition of SC improves microbial activity, optimizes microbial community structure, promotes the expression of c-type cytochromes, and is conducive to the formation of electroactive biofilms. This study not only describes a technique for the preparation of high-performance and low-cost anodes, but also sheds some light on the rational utilization of waste resources such as aerobic activated sludge.

## 1. Introduction

Microbial fuel cells (MFCs) can convert chemical energy in organic matter into electric energy by using the oxidation and metabolism mechanism of anaerobic bacteria on anodic electroactive biofilms (EABFs) [1,2]. Compared with other electrochemical cells, such as liquid flow cells and ordinary fuel cells, MFCs do not need external energy input and have high energy conversion efficiency [3,4]. MFCs can use all biodegradable organics and wastewater as fuels [2]. In wastewater treatment, MFCs can generate electrical energy while efficiently treating wastewater, which not only significantly reduces the operation cost of sewage treatment plants, but also makes efficient use of waste resources [3]. Traditional wastewater biological treatment approaches mainly involve anaerobic digestion and aerobic treatment technologies [5]. Anaerobic digestion technology is mainly applicable to high concentration wastewater [6,7]. Although it can produce fuel gases such as methane or hydrogen from organic wastewater, it also produces gases with no practical value such as carbon dioxide, hydrogen sulfide and nitrogen [5,8]. The energy utilization mode is complex and has significant requirements for production conditions. Aerobic treatment technology is mainly applicable to medium and low concentration wastewater. With large sludge production, aeration is required to maintain oxygen concentration, and the cost of sludge disposal and aeration is high [8]. MFCs have a wide application range, low sludge output and directly utilize electric energy. Thus, MFCs have incomparable technical advantages compared with traditional wastewater biological treatment technologies [3,7]. Therefore, the application of MFCs in wastewater treatment has broad prospects. However, recently, it was reported that the maximum current and power densities of MFCs are 3.85 mA·cm^2^ and 0.66 mW·cm^2^, respectively, and the power production is still relatively low, i.e., 2–4 orders of magnitude below that of chemical fuel cell [9]. In addition, the cost of MFCs is also relatively high, notably due to the commonly used proton exchange membrane and cathode platinum catalyst, which greatly increase the cost [10]. Thus, practical applications for microbial fuel cells are limited.

The anode, as the core of MFCs, plays a key role in the extracellular electron transfer (EET) between electroactive microorganisms (EAMs) and electrodes [8]. The properties of the anode materials significantly affect the growth of EABFs. An ideal anode material should have high biocompatibility and conductivity, cost efficiency, and easy commercial production [11]. Traditional carbon-based materials, such as carbon cloth (CC), carbon felt, carbon paper and graphite rod, have been widely used in MFCs. However, due to the low electrochemical activity of carbon-based materials, their application in MFC is limited [12]. Therefore, it is necessary to modify carbon-based materials to improve the power generation performance of MFCs. Given the poor durability of carbon paper [4], the high resistance of carbon felt [11] and the smooth surface of graphite rods [13], CC is often used to modify carbon-based materials because of its good durability, low resistance and easy modification [14]. Carbon-based nanomaterials, such as graphene, carbon nanotubes, conductive polymers and carbon nanoparticles have been widely used to modify carbon-based electrodes because of their good electrocatalytic activity and conductivity [11,12]. Liu et al., modified CC with tungsten carbide nanoparticles, which significantly improved the electrocatalytic performance of the resulting electrode. The power density reached 3.26 W·m^−2^, which was 2.14 times that of the bare CC anode (1.52 W·m^−2^) [15]. Li et al., modified a CC with polydopamine (PDA) and graphene oxide (rGO), which improved the hydrophilicity and significantly reduced charge transfer resistance. Its power density reached 2.047 W·m^−2^, which was 6.1 times that of the bare CC electrode [10]. Liu et al., modified CC with graphene to enhance its biocompatibility. The power density and energy conversion efficiency were 2.7 times and 3 times higher than those of the bare CC electrode, respectively [16]. The above studies show that the modification of CC anodes can significantly improve the performance of MFCs. However, the preparation process of these nano materials is complex and yields are low [11]. It is estimated that the preparation cost of these nanomaterials is high, accounting for more than half of the total cost of MFCs [17]. Therefore, reducing the cost of MFC anode materials while maintaining excellent performance is very important for further practical applications of MFCs.

Recently, carbon materials derived from natural biomass materials have attracted extensive interest because of their low cost and sustainable resource utilization. Porous carbon derived from biomass wastes such as almond shells, pomelo peels, towel gourds, kenaf stems, silkworm cocoons and chestnut shells have been used to produce the anodes of MFCs, showing excellent power generation performance; such approaches also open up a new path for the utilization of natural waste [18,19,20]. Sewage sludge, a byproduct of wastewater treatment, is abundant but remains expensive to dispose of. According to statistical reports, China produces approximately 11.2 million tons of dry sludge every year, while the countries of the European Union produce 10.0 million tons [21]. The cost of sludge treatment and disposal is approximately 60% of the total operating cost of sewage treatment plants [22]. In addition, the sludge contains pathogenic bacteria, organic pollutants, heavy metals and other harmful substances, and its improper treatment can be harmful to human health and the environment. Compared to incineration, landfill, anaerobic digestion, and composting, pyrolysis treatment can effectively reduce the toxicity of dry sludge, yielding sludge carbon (SC) and high value-added fuel [23]. It is considered a safe, stable, and low-cost sludge treatment method. Previously, researchers have applied SCs prepared by one-step pyrolysis to lithium-ion batteries, supercapacitors, and environmental catalysts [23]. Sludge-derived carbon has been shown to be feasible as a low-cost conductive material. However, the effects of the physical and chemical properties of SC (such as specific surface area, porosity, functional groups and metal phase structure) on MFC anodes, especially on extracellular polymers (EPS) in anode biofilms, and the mechanisms of electron transfer, are still unclear. As an electrode material, SC still poses scientific and technical challenges.

The purpose of this study was to investigate the effects of SC characteristics on the EABFs of MFC anodes. The parameters assessed included power generation, wastewater treatment ability, electrochemical performance, EPS secretion, microbial activity, diversity, and metabolic pathways. It aimed to provide a theoretical basis for the practical application of sludge carbon in MFCs, and to lay out a safe and effective method for the utilization of sludge resources.

## 2. Materials and Methods

### 2.1. Preparation of Electrodes

The anode and cathode were made of carbon cloth, purchased from WENTE Co., Ltd., Nanjing, China. The CC for the anode substrate (1.0 × 2.0 cm^2^) was relatively hydrophilic, with a thickness of 0.36 mm and a resistivity of less than 5 mΩ·cm^2^. In order to increase the hydrophilicity of the anode carbon cloth, the treatment method was as follows: 10% nitric acid solution and 10% sulfuric acid solution were mixed in a 3:1 ratio; then, the carbon cloth was soaked in this solution for 12 h [15]. The CC for the cathode substrate was hydrophobic, with a thickness of 0.41 mm and a resistivity of less than 13 mΩ·cm^2^.The preparation process of the anode was as follows: Aerobic activated sludge was collected from a sewage treatment plant in Nanning, China. The sludge was dried at 60 °C for 24 h and broken by grinding. It was then pyrolyzed at 600, 800, 1000, and 1200 °C for 2 h to obtain SC powder, filtered, and dried to obtain SC600, SC800, SC1000 and SC1200 samples. The obtained sludge carbon samples (1.5 mg cm^−2^) were sprayed onto a carbon cloth to obtain SC600/CC, SC800/CC, SC1000/CC and SC1200/CC electrodes. Bare carbon cloth (CC) (1.0 × 2.0 cm^2^) was used as the control electrode. The cathode was prepared to use activated carbon as the catalyst, 5 wt% Nafion solution as the binder, and carbon cloth as the collector [24]. The effective working area of the cathode CC was 4.91 cm^2^.

### 2.2. MFC Construction

For air cathode microbial fuel cells, the elimination of the proton exchange membrane can significantly reduce the power output and cost of the cell. The reactors used in this study were 100 mL single-chamber membrane-free air cathode bottle MFCs. The MFC was inoculated on a super clean workbench. Firstly, 10 mL of pre-acclimated anaerobic granular sludge was sucked into the anode chamber with a disinfection syringe. Then, 80 mL of prepared artificial wastewater was sucked. The artificial wastewater was mainly composed of sodium acetate (1.5 g·L^−1^), phosphoric acid buffer solution (50 mM), trace elements (12.5 mL·L^−1^), and vitamin solution (5 mL·L^−1^) [24]. The formula of the nutrient solution is shown in the Appendix A. Then, an external resistance of 1000 Ω was employed to connect the electrodes [24]. Intermittent water inlet mode was applied to the MFCs. The anode medium was replaced when the cell voltage passed below 50 mV. All experiments were conducted in triplicate, and average values were calculated. The external voltage generated by the MFCs was measured using a data acquisition instrument (Keithley6510, Cleveland, OH, USA). The polarization and power density curves were measured by gradually changing the external resistance (2000 Ω to 80 Ω) [7]. According to the formulas I = U/R and P = UI, the current and power under the corresponding resistances were calculated, respectively [25]. Cyclic voltammetry (CV) tests were performed in a three-electrode system using an electrochemical workstation (Chenhua, Shanghai). The prepared anode, a saturated calomel electrode (SCE), and a platinum wire were used as the working, reference and counter electrodes. The measurement range was −0.6~0.6 V, the scanning speeds were (1, 5, 10, 15 and 20 mV/s), and the static time was 2 s [9].

### 2.3. Materials Characterizations

The surface morphology and properties of the samples were characterized by scanning electron microscopy (SEM-EDX, SU8020, Hitachi hi tech, Tokyo, Japan), high-resolution electron microscopy (HR-TEM, FEI TECNAI G2 F30, FEI NanoPorts, Hillsboro, OR, USA), X-ray powder diffraction (XRD, RIGAKU D/MAX 2500V, Japan Science Corporation, Tokyo, Japan), X-ray photoelectron spectroscopy (XPS, ESCALAB 250XI+, Thermo Fisher Scientific, Waltham, MA, USA) and Raman spectroscopy (Raman, InVia Reflex, Renishaw, London, UK). The pore size distribution and specific surface area of the samples were measured by Barrett-Joyner-Halenda (BJH) method and Brunauer-Emmett-Teller (BET) theory (NOVA4200E, Cantar Instruments, Boynton Beach, FL, USA). The surface functional groups of the samples were characterized in the range of 4000–300 cm^−1^ by Fourier transform infrared spectroscopy (FTIR, TENSOR II, Bruker, Karlsruhe, Germany). A Diamond TG/DTA instrument (DTG-60(H), Hitachi hi tech, Tokyo, Japan) was used for a thermogravimetric (TG) analysis of the samples. Confocal scanning electron microscopy (CLSM, LSM800, zeroK Nanotech, Gaithersburg, MA, USA) was used to characterize the cell activity of the EABFs. Chemical oxygen demand (COD) was detected using spectrophotometry and a multiparameter water quality analyzer (Lianhua Technology Co., Ltd, Shanghai, China). The total suspended solids (TSS) in the anaerobic granular sludge was calculated by drying and weighing at 105 °C, while the volatile suspended solids (VSS) were calculated by calcination at 600 °C. The zeta potential of the samples was characterized using a zeta potential particle sizer (Nano-ZS90X, Marvin Instrument Equipment Co., Ltd, London, UK).

### 2.4. EPS Extraction and Analysis

Extracellular polymers (EPS) on anode biofilms were extracted by the water bath heating method [15]. After power generation, the CC was cut with sterile scissors and placed into centrifuge tubes containing 0.9% sodium chloride solution, rotated for 10 min, and centrifuged at 4000× *g* rpm for 15 min. The supernatant was discarded, replenished with sodium chloride solution, the above operations were repeated, and vibrated in an 80 °C water bath shaker for 30 min. Finally, the supernatant was centrifuged at 9000× *g* rpm for 10 min and filtered through a 0.22 μm membrane to obtain tightly bound EPS. The contents of polysaccharides, humic substances, protein, and outer membrane c-type cytochromes (OM c-Cyts) in EPS were detected using UV/visible spectrophotometer (Agilent 8453, Agilent Technology Co., Ltd., Santa Clara, CA, USA). EPS was characterized using a three-dimensional excitation-emission matrix (3D-EEM) fluorescence spectrometer (Hitachi F-7000, Hitachi hi tech, Tokyo, Japan). The main test parameters were as follows: the scanning ranges of the excitation spectrum (Ex) and emission spectrum (EM) were 220–500 nm and 220–550 nm, respectively, while the scanning step was 5 nm.

### 2.5. Microbial Community Analysis

The anodes were collected after power generation, and the genomic DNA was extracted from the biofilm samples using the E.Z.N.A.^®^ Mag-Bind Soil m DNA Kit (OMEGA) according to the manufacturer’s instructions [25]. The V3-V4 region of the 16S rRNA gene was amplified using polymerase chain reaction (PCR) with primers 338F (5′-ACTCCTACGGGAGGCAGCA-3′) and 806 R (5′-GGACTACHVGGGTWTCTAAT-3′) [26]. The same loading buffer volume was mixed with the PCR product, and electrophoresis was performed on a 2% agarose gel. The products were purified using a pre-QUS kit™. The PCR products were detected and quantified using a fluorometer. The NEXTFLEX Rapid DNA-Seq kit was used to build the libraries. After purification and quantification, the samples were tested for 16S rRNA gene sequencing based on the Illumina MiSeq platform. Based on the Kyoto Encyclopedia of Genes and Genomes (KEGG) database, the macrogenomic information of microbial metabolic function was predicted using the PICRUSt pipelines program [9].

## 3. Results

### 3.1. Characterization of Material

Figure 1 shows that the morphological changes in SC were significantly related to the carbonization temperature. The SC600 and SC800 samples showed disordered sheet structures. For the SC1000 sample, the surfaces were relatively rough, and carbon microspheres and small pores appeared due to the release of volatile substances and the sintering effect, which converted a large number of inorganic parts into mineral-like compounds and induced the carbon phase to warp some metals [27]. For SC1200 samples, high temperature enhanced the shrinkage of carbon, hindered the development of pores, led to the collapse or deformation of coke, and reduced the pore volume. Furthermore, it can be observed from the TEM images (Appendix A) that the morphology of SC changed significantly with the increase of pyrolysis temperature, with carbon particles appearing in SC1000 and SC1200 samples, which is consistent with the SEM results.

As shown in Table 1, the pH, zeta potential, and ash content of sludge carbon was linearly related to the carbonization temperature. Increasing the pyrolysis temperature can increase the zeta potential of the sludge carbon surface. When the carbonization temperature was low, the measured zeta potential was negative because the original sludge was rich in negative groups such as carboxyl and hydroxyl groups, and the sludge carbon surface was negatively charged [21]. When the carbonization temperature was 1000 °C or above, the zeta potential was positive and the mud carbon surface was positively charged. This was because the oxygen-containing functional groups with negative charge were basically decomposed, and a large number of amino and metal groups were exposed on the carbon surface of the sludge [28]. The electrical properties of the material surface changed from negative to positive, which is useful for the electrical absorption of micro-organizations with a negative charge. Another important feature of an ideal anode is high conductivity. The results showed that the SC1000 sample had the highest conductivity (57.43 s·m^−1^). In summary, the SC1000 sample may be the most suitable anode material. Its high conductivity and low absolute zeta potential value enable the adhesion of EAMs and the formation of EABFs. As shown in Appendix A, TGA showed a weight-loss temperature range of 200–1300 °C, indicating that sludge carbonization began at 200 °C and mainly ended at 1300 °C, and the mass was reduced by 51%. As shown in Figure 2, The FTIR spectra of all samples showed the following stretching vibrations: -OH of ethanol with a peak at 3450 cm^−1^, C-H at 2849 cm^−1^, N-H at 1650 cm^−1^, unsaturated aldehyde with C=O at 1406 cm^−1^, C-O of amine at 1100 cm^−1^, and C-H at 789 cm^−1^ and 2928 cm^−1^ [21]. The raw sludge was rich in carboxyl and hydroxyl groups [21]. As the carbonization temperature increased, the intensity of these peaks gradually decreased. This was due to the decomposition of oxygen-containing functional groups. Because of their negatively charged surface, oxygen-containing functional groups are not favorable for bacterial adhesion, thus affecting the electron transfer path [25].

As shown in Figure 3 and Appendix A, the specific surface area and pore structure of SC were significantly affected by the carbonization temperature, which greatly influenced the catalytic activity of EABFs. The N_2_ adsorption/desorption curves of the SC samples belonged to type IV isothermal curves, and there were notable H_3_ hysteresis curves (Figure 3a). This adsorption hysteresis phenomenon is related to the mesoporous and microporous characteristics of the slit [29]. Appendix A summarizes the BET and pore-size distributions of the SC samples. The results showed that the specific surface areas of the SC samples gradually increased as the carbonization temperature increased. The specific surface area of the SC1000 sample was the largest (216.00 ± 0.005 m^2^·g^−1^). However, when the carbonization temperature was further increased to 1200 °C, the specific surface area of the sample decreased (176.55 ± 0.010 m^2^·g^−1^). The pore size distribution of the SC was further studied (Figure 3b and Appendix A). There were two types of pores in all SC samples: micropores and mesopores, with an average size of approximately 8 nm. The total pore volume and mesopore number of the SC800 sample were the largest, while the total pore volume and mesopore volume of the SC1000 sample reduced, but the micropore volume increased, which may have been due to the collapse of some mesopores under high-temperature conditions, resulting in pore blockage [30]. It could be predicted that the special mesoporous and microporous structures produced on the surface of SC may provide active sites for the catalytic reaction and enhance the electrocatalytic ability, thus enhancing proton transfer and charge transfer [15,25,31]. The increase in the specific surface area also has a considerable effect on the decrease in the internal resistance [31]. In short, the SC1000 sample had a good specific surface area and rich pore structure for proton and electron transfer, substrate transport, and biofilm formation in MFCs [11,25,29].

As shown in Figure 3c, an XRD analysis of the structure and crystallinity of the SC samples at varying carbonization temperatures indicated pyrolysis differences in the samples at different temperatures. The characteristic diffraction peaks at 26.3° and 41.0° correspond to the (002) and (101) lattice planes of graphite carbon, respectively, confirming the existence of graphite carbon. The characteristic diffraction peaks at 40.9° and 54.1° corresponded to the (110) and (116) lattice planes of α-Fe_2_O_3_ (JCPDS. Card 33-0664). The characteristic diffraction peak at 35.4° corresponded to the (111) lattice plane of Fe_3_O_4_ (JCPDS. Card 72-2303). The characteristic diffraction peaks of Fe_3_C (PDF No. 01-089-3689) are 43.7° and 54.7° [29,32]. As the carbonization temperature increased, the intensities of α-Fe_2_O_3_ and Fe_3_O_4_ decreased, while the intensities of graphite carbon and Fe_3_C increased. The existence of graphite carbon and Fe_3_C supported the conductivity of the SC [29]. The graphitization degree of SC was studied by testing the carbonized samples at different temperatures using Raman spectroscopy (Figure 3d). The SC samples showed two characteristic strong peaks at 1350 cm^−1^ and 1599 cm^−1^, corresponding to the defect D and G peaks of graphite, respectively [25]. The strength ratios (I_D_/I_G_) of the D and G bands decreased as the carbonization temperature increased, indicating that the graphitization degree of SC improved. In addition, compared to the SC600 and SC800 samples, the SC1000 and SC1200 samples had wider peaks at 1350 cm^−1^, indicating that the nanoparticles in the sludge carbon samples were smaller.

EDX was performed to detect the types and distribution of elements in SC1000 (Appendix A). According to the EDX results, SC1000 samples were rich in elements. In addition to the main elements, i.e., C, N, O, and Si, these samples also had low contents of metals such as K, Ca, Mg, Al, Fe, Cu, Zn, Ti, as well as trace amounts of P and S; these heteroatoms were evenly distributed. Among them, C, N, P, and S are conducive to enhancing the graphite properties and hydrophilicity of SC [25]. Al, Ca, and Mg are useful for the formation of biochar metal skeletons [23]. It is speculated that transition metal elements such as Fe, Ti, Cu and Zn can provide rich active sites for redox reactions and improve the electrocatalytic activity of EAMs [28,29]. The specific content of each element in the SC samples was determined using XPS (Appendix A). Interestingly, with the increase in carbonization temperature, the contents of the other elements continued to decrease, except for C and S. This was mainly due to the loss of volatile substances during pyrolysis. In addition, with the increase in carbonization temperature, the contents of oxygen- and nitrogen-containing functional groups decreased, which was consistent with the FTIR results. It is well known that the reduction of oxygen content and pyridine groups enhance electron transfer [25,33]. The types and contents of the elements in the sludge carbon were further confirmed by XPS. Appendix A shows the XPS full spectrum results of the SC samples carbonized at 600, 800, 1000, and 1200 °C. According to the results, the elements in the SC samples mainly comprised C, N, O, P, Si, Ca, Mg, Al, Fe, and P; this was consistent with the EDX test results.

Furthermore, changes in the chemical forms of the main elements, i.e., C, N, and Fe, in the SC samples prepared at different carbonization temperatures were analyzed by XPS. As shown in Figure 4, the XPS high-resolution spectrum of C 1s mainly included C-C and C=C graphite carbon with a binding energy of 284.80 eV, C-N with a binding energy of 285.78 eV, C-O with a binding energy of 286.88 eV and C=O with a binding energy of 289.12 eV [23]. It was further indicated that with the increase in carbonization temperature, the oxygen-containing functional groups gradually decreased, and most amorphous carbon was transformed into SP^2^ graphite carbon and C-N. Graphite carbon improves the conductivity of electrodes, and the C-N bond could increase its hydrophilicity, reduce charge transfer resistance, and facilitate bacterial adhesion [10,12,31,32]. The XPS high-resolution spectra of N 1s mainly indicated pyridinic-N with a binding energy of 398.80 eV, graphitic-N with a binding energy of 401.50 eV, Fe-N with a binding energy of 399.60 eV, oxidized-N with a binding energy of 402.90 eV and pyrrolic-N with a binding energy of 400.69 eV [29,32]. When the carbonization temperature was lower than 1000 °C, the proportion of pyridinic-N and oxidized-N decreased, while the contents of graphitic-N, pyrrolic-N, and Fe-N increased as the temperature progressed. It is well known that graphitic-N can improve the conductivity of materials and accelerate electron transfer, and pyrrolic-N can improve the electrochemical reaction rate. Fe-N improves the electrocatalytic activity of EABFs and accelerates electron transfer [34]. The XPS high-resolution spectrum of Fe 2p showed that the oxidation state of Fe on the surface of the SC sample was complex, and that the difference was significant with the change of carbonization temperature. Due to the presence of various iron species, sludge carbon exhibits ferromagnetism, which is conducive to microbial adsorption [23]. Particularly, for the SC1000 sample, the presence of a Fe-C binding site with a binding energy of 720.7 eV confirmed the existence of Fe_3_C. The surface of the CC anode was smooth, which was unfavorable for microbial adsorption (Appendix A). The surfaces of CC loaded with sludge carbon were relatively rough, which increased the microbial contact area. For SC1000/CC, the adhesion between the sludge carbon and CC was closer, which was better for microbial adsorption (Appendix A).

### 3.2. Electrode Electrocatalytic Activity

CV analyzed the redox medium composition and redox potential of biofilms. All CV curves showed a typical S-type anode catalyst curve with sodium acetate as the substrate (Figure 5). This indicated the formation of electroactive biofilms and showed that all MFC systems may adopt similar electron transfer paths. However, except for the CC anode, the CV curves of other SC anodes showed a pair of redox main peaks, i.e., mainly redox pairs centered on −0.38 V (cathode) and−0.06 V (anode). The midpoint potential was−0.2 V, which is within the electron transfer activity range of outer membrane c-type cytochromes (OM c-Cyts), indicating that the electron transfer path between the biofilm and the electrode is mainly short-range direct electron transfer (DET) mediated by OM c-Cyts [35,36,37]. The capacitance area of the SC anodes was much larger than that of the CC anode. The above results show that the EABFs on SC electrodes have high electrocatalytic activity. In addition, the peak current density and capacitance area of the anodes revealed a significant linear relationship with the sludge carbonization temperature, and increased with increasing scanning speed.

### 3.3. MFC Performance

As shown in Figure 6a, in the first six cycles (approximately 25 days), the COD removal rates of all MFCs showed an increasing trend, and after the sixth cycle of power generation, the COD degradation rate stabilized. There was a correlation between COD removal efficiency and carbonization temperature. MFCs equipped with SC1000/CC had the largest COD removal efficiency (97.63 ± 0.039%), i.e., higher than that of CC (90.97 ± 0.035%). In addition, during stable power generation, the VSS/TSS of the granular sludge in MFCs equipped with SC electrodes was significantly higher than that of MFCs equipped with CC (Figure 6b), and the value of VSS/TSS increased with the increase of sludge carbonization temperature. This indicated that sludge carbon can increase the organic components in granular sludge, thus improving the microbial biomass and microbial activity. Interestingly, the trend regarding the COD removal efficiency of the MFC reactor was consistent with that of VSS/TSS. This may have been because the addition of SC promoted the enrichment of microorganisms, stimulated microorganisms to secrete EPS, and accelerated the carbohydrate metabolism of exoelectrogens and methanogens, thereby accelerating the degradation rate of organic matter [5]. Additionally, carbonization temperature also strongly affected the internal resistance, power density and current density of SC anodes (Figure 6c,d). According to the polarization curve, there was a significant difference in anode open circuit voltage. The open circuit voltage of the SC anode was 0.640 V (SC600/CC), 0.686 V (SC800/CC), 0.725 V (SC1000/CC) and 0.698 V (SC1200/CC), i.e., higher than that of the CC anode (0.626 V). Under the same voltage conditions, the current density of SC1000/CC was the largest, indicating that SC1000/CC has the highest electrochemical oxidation activity. The SC1000/CC anode generated a maximum areal power density of 2.165 ± 0.021 mW·m^−2^ and current density of 5.985 ± 0.015 A·m^−2^, which was 3.017- and 2.992-fold that of the CC anode. The SC1000/CC anode had lower internal resistance and produced higher current density and power density, which meant the electroactive biofilm had higher electrocatalytic activity. As shown in Figure 7, the sludge carbonization temperature significantly affected the voltage output of the MFC system. Compared to the MFCs equipped with CC electrodes, the output voltage of the MFCs equipped with SC electrodes significantly improved. Among them, the MFCs equipped with SC1000/CC anodes showed the best power generation performance, with a maximum closed-circuit voltage of 0.501 V and an average power generation cycle of 146 h. The maximum closed-circuit voltage of the MFC equipped with CC anodes was only 0.323 V, and the average power generation cycle was 100 h.

### 3.4. Anode Biofilm Characterizations

As shown in Appendix A, in the process of power generation, the modification of SC significantly promoted the secretion of EPS, further confirming that SC can improve microbial biomass and microbial activity. It is well known that in the biofilm-producing current, c-Cyts play a key role in interspecific electron transfer and extracellular electron transfer [36]. It is believed that the higher the concentration of c-Cyts, the higher the electron transfer efficiency between EAMs and electrodes [35]. Some studies have shown that the c-Cyts in EPS can closely bind to the active sites of transition metals Fe (III), Cu (II), and Zn (II) to promote electron transfer [23,38]. It can be seen in Figure 8a–f that in all MFCs, the absorbance of c-Cyts at a wavelength of 419 nm in the anode biofilm gradually increased with power generation, and that the addition of sludge carbon at different carbonization temperatures changed the absorbance of c-Cyts, indicating that sludge carbon affected the secretion of c-Cyts and further influenced the power generation. SC1000/CC had the highest absorbance at 419 nm, indicating that its c-Cyts concentration was the highest, so it can continuously absorb electrons from the embedded bacteria and transfer them to the electrode. As shown in Figure 8g–l, EPS extracted from the anode biofilm was analyzed using EEM fluorescence spectroscopy. Among the observed peaks, peak A (Ex/Em = 280–295/320–335 nm) was tryptophan-like acid and soluble microbial by-products, Peak B (Ex/Em = 380/435–450 nm) was humic-like acid, and peak C (Ex/Em = 420–425/450–470 nm) was coenzyme F_420_, which plays an important role in the hydrogen nutrition pathway and is related to the methanogenic metabolic activity [5,39]. Peak D (Ex/Em = 230/325–330 nm) was protein-like. Compared to the inoculated sludge, the intensity of peaks A and B in the MFC anode biofilm gradually increased, and peak C appeared. Compared to CC, the intensities of peaks A, B, and C in the SC anode biofilms gradually increased, and peak D appeared. This was consistent with the conclusion in Appendix A, which further proved that the SC can change the composition and content of EPS. As shown in Appendix A, SC significantly promoted the increase in microbial biomass in the biofilm and the secretion of EPS. The EPS matrix covered carbon defects on the electrode surface and strengthened cell adhesion. As shown in Appendix A, SC-modified CC can improve microbial activity and promote the growth of EABFs. The ratio of live cells on the SC1000/CC anode biofilm was the largest, indicating that the anode had high biocompatibility.

### 3.5. Microbial Community Analysis

The α diversity represents the microbial community diversity on the anode EABFs. As shown in Appendix A and Appendix A, compared to the CC anode biofilm, the biofilm community diversity, richness, and the total number of species in the SC anode biofilms increased. These factors were linearly related to carbonization temperature, indicating that the SC has a significant impact on the community diversity of the anode biofilm. The effect of the SC characteristics on the microbial community structure of EABFs was further explored by high-throughput sequencing (Figure 9). SC significantly affected the community structure of the anode biofilm, thus directly affecting the power generation of MFCs. As shown in Figure 9a, at the phylum level of archaea, Euryarchaeota, and halobacteria were dominant. Among them, Euryarchaeota represent an irreplaceable functional microorganism in the anaerobic digestion process [40], and their numbers increased with an increase in the sludge carbonization temperature. In contrast, the content of halobacteria decreased with an increase in the sludge carbonization temperature, indicating that SC changes the structure of archaea, and that the characteristics of SC will selectively enrich functional microorganisms. At the genus level of archaea (Figure 9b), Methanobacterium and Methanosaeta were dominant. They can participate in EET and interspecific direct electron transfer (DIET) [5,39,40]. Methanosacrina is an obligate anaerobic bacterium that was significantly enriched in the SC1000/CC anode. It can possibly perform an EET. Hence, SC optimizes the archaeal community structure, selectively enriches functional microorganisms, and enhances the hydrogen methane production pathway.

As shown in Figure 9c, at the phylum level, the microbial communities on all anodes were similar, but the relative abundance of dominant bacteria was significantly different. Compared to the CC anode, the relative abundance of Proteobacteria, Firmicutes, and Bacteroidetes in the SC anodes decreased, and there was a correlation with the carbonization temperature. In contrast, the relative abundance of Desulfobacterota, Synergistota, and Actinobacteria increased. These results showed that SC could promote a synergistic effect between exoelectrogens and Synergistota. Desulfobacteria and Actinobacteria are related to organic matter degradation and usually control the acetate oxidation community [41], which contains a large number of electrochemically-active species [2,42]. The relative abundance of Desulfobacterota on the SC1000/CC biofilm was the highest (35%), indicating that the biofilm was rich in electroactive bacteria.

As shown in Figure 9d, at the genus level, SC increased the diversity of microbial community. The dominant bacteria in each MFC were Geobacter and Lentimicrobium. Geobacter is a common exoelectrogen that can secrete cytochromes and participate in direct electron transfer [35,37,42]. This may be reflected in the power generation performance of MFC systems. Direct electron transfer may occur between Geobacter and methanogens, which can cooperate in power generation [43]. Lentimicrobium is a strictly anaerobic gram-negative bacterium. Studies have shown that it may form a consortium with other EAMs to convert acetate into electric energy [42]. Interestingly, SC significantly increased the relative abundance of Thermovirga, which was positively correlated with carbonization temperature, and may produce cytochrome to participate in EET [37]. In contrast, the relative abundance of Pseudomonas was significantly reduced due to the presence of SC, which promoted EET by secreting phenazines [42]. The results showed that the electron transfer path of SC anode biofilms was mainly short-range DET of c-type cytochromes, which was consistent with the data presented in the CV curves in Figure 5. The above results show that the microbial community structure of EABFs is directly related to the characteristics of the anode materials, and that there may be electron transfer between Archaea and EAMs. The metabolic pathways of microorganisms determine the flow of electrons and protons, which affect the performance of electricity production. In metabolism clusters, all reactors used amino acid, carbohydrate, and energy metabolism as the main metabolic pathways (Appendix A). The SC1000/CC anode biofilm had the highest amino acid, carbohydrate, energy, and lipid metabolism. High carbohydrate metabolism indicates that microorganisms decompose organic matter quickly, which is related to increased output voltage [43].

## 4. Conclusions

The sludge carbon prepared by one-step pyrolysis had the advantages of high conductivity, good pore structure, high biocompatibility, high carbon content, rich heteroatom composition, and low cost. It significantly improved the activity and diversity of microorganisms on the anode biofilm, optimized the composition of the microbial community, regulated the metabolic pathway of microorganisms, promoted the secretion of EPS and the expression of cytochrome, and strengthened the electron transfer ability of EABFs. Therefore, an anode processed at 1000 °C generated a maximum areal power density of 2.165 ± 0.021 W·m^−2^ and current density of 5.985 ± 0.015 A·m^−^^2^, which was 3.017- and 2.992-fold that of the CC anode. This study provides a theoretical basis for the practical application of sludge carbon in MFCs and provides a new direction for the rational utilization of biomass waste resources.

## Figures and Tables

**Figure 1 membranes-12-00120-f001:**
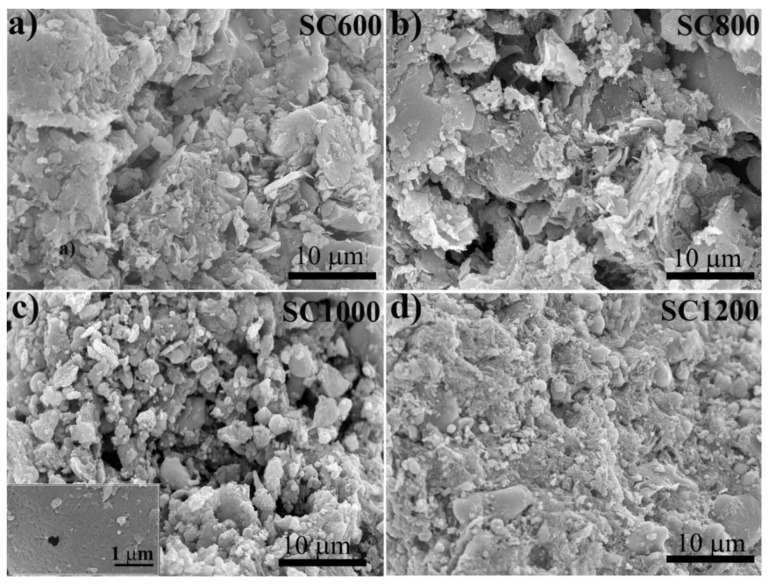
SEM images of SC samples under different carbonization temperature: (**a**) SC600, (**b**) SC800, (**c**) SC1000, and (**d**) SC1200.

**Figure 2 membranes-12-00120-f002:**
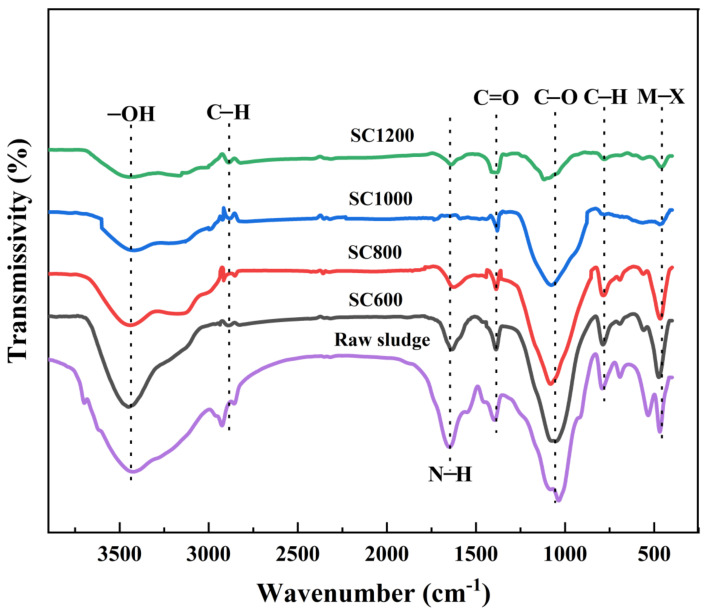
FTIR spectra obtained from raw sludge and SC samples.

**Figure 3 membranes-12-00120-f003:**
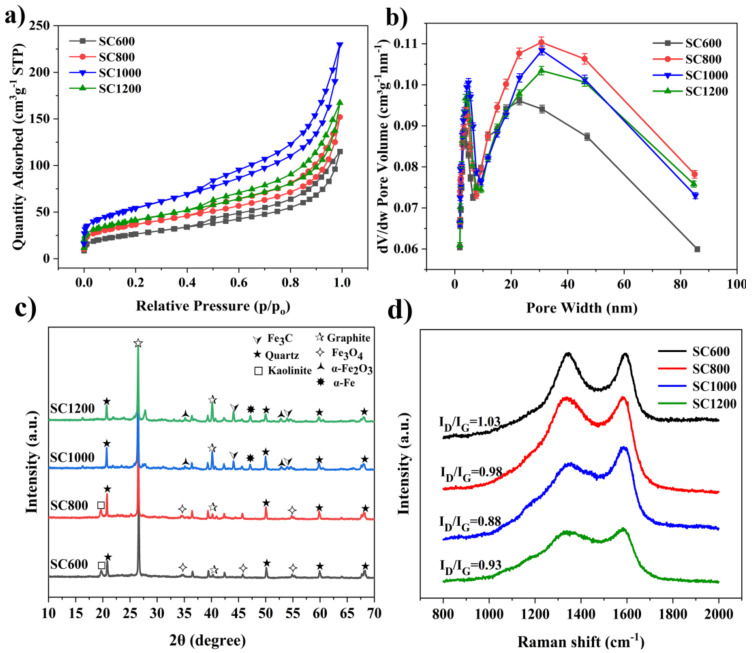
(**a**) N_2_ adsorption-desorption isotherms, (**b**) pore size distribution, (**c**) powder XRD patterns and (**d**) Raman spectra of the SC samples.

**Figure 4 membranes-12-00120-f004:**
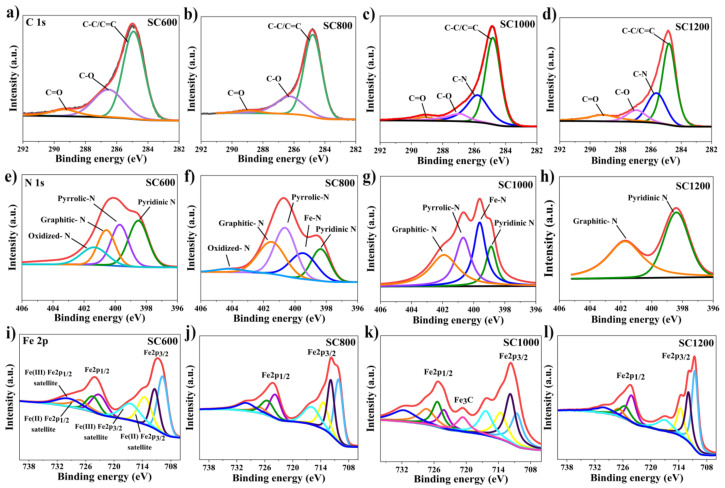
C1s spectra for (**a**) SC600, (**b**) SC800, (**c**) SC1000 and (**d**) SC1200; N1s spectra for (**e**) SC600, (**f**) SC800, (**g**) SC1000 and (**h**) SC1200; Fe2p spectra for (**i**) SC600, (**j**) SC800, (**k**) SC1000 and (**l**) SC1200.

**Figure 5 membranes-12-00120-f005:**
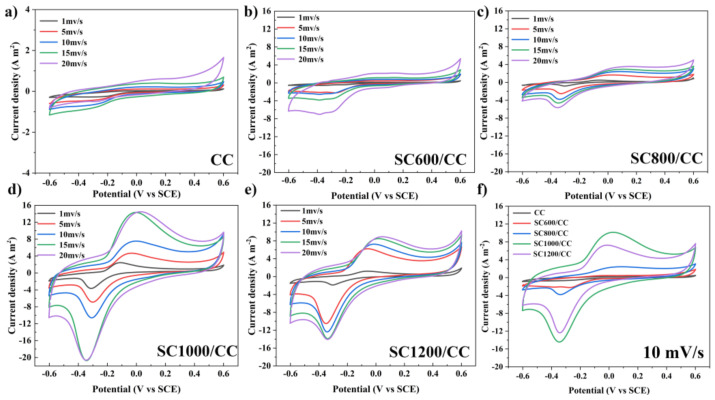
CV plots of (**a**) CC, (**b**) SC600/CC, (**c**) SC800/CC, (**d**) SC1000/CC, (**e**) SC1200/CC with 1, 5, 10 and 15 mV/s scanning rates and (**f**) CV plots of the electrodes with 10 mV/s scanning rates at three-electrode electrochemical systems after 54 d of batch mode operations.

**Figure 6 membranes-12-00120-f006:**
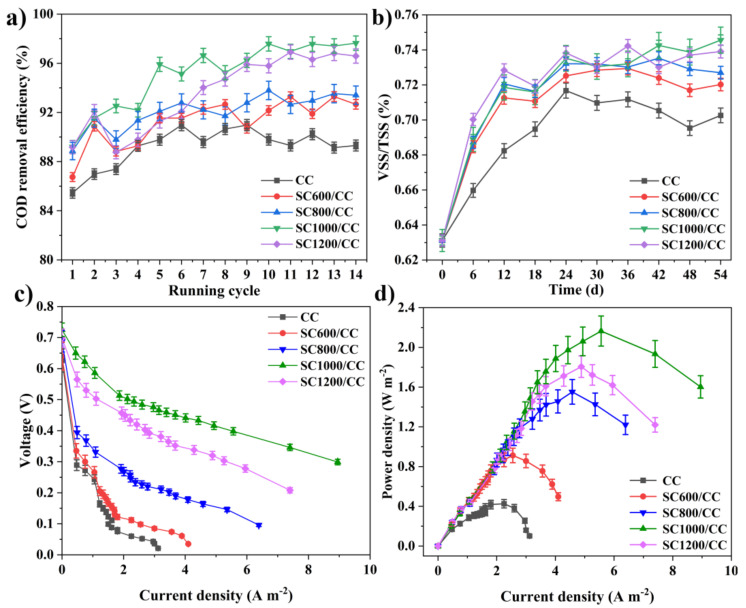
(**a**) COD removal efficiency of MFCs, (**b**) VSS/TSS of inoculated anaerobic granular sludge at MFC systems, (**c**) polarization curves, and (**d**) power density of MFCs.

**Figure 7 membranes-12-00120-f007:**
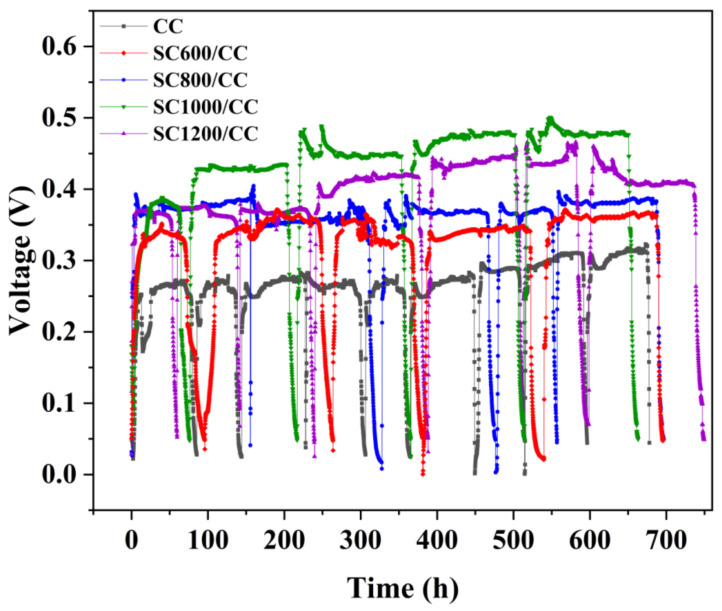
Reproducible cycles of output voltages produced in MFCs equipped with different anodes.

**Figure 8 membranes-12-00120-f008:**
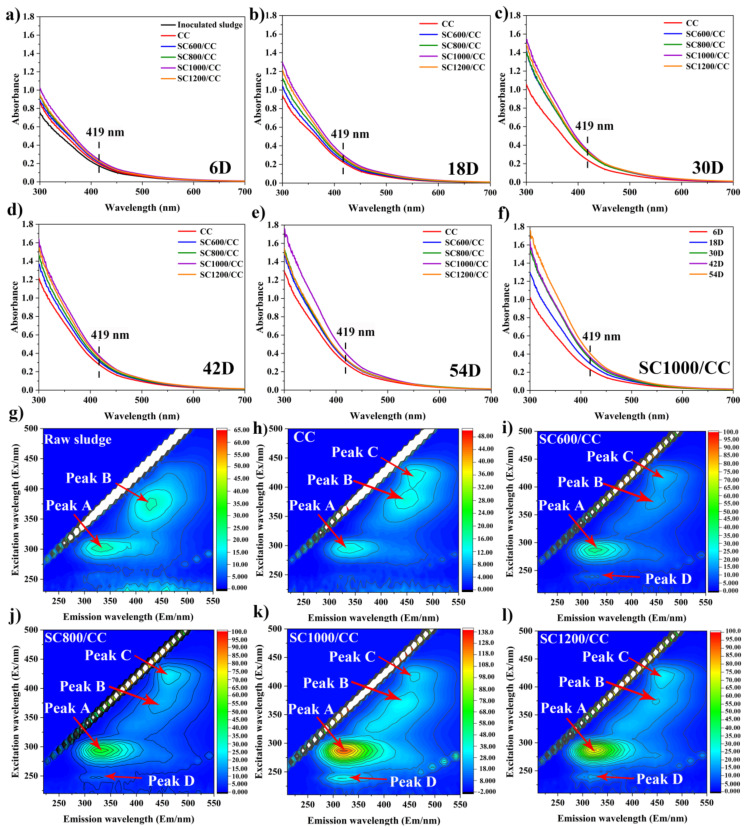
Absorbance of c-Cyts for inoculated sludge and different anodic biofilms at MFCs on (**a**) 6D, (**b**) 18D, (**c**) 30D, (**d**) 42D, and (**e**) 54D. Absorbance of c-Cyts for (**f**) SC1000/CC. EEM fluorescence spectra of EPS extracted from (**g**) inoculated sludge, (**h**) CC, (**i**) SC600/CC, (**j**) SC800/CC, (**k**) SC1000/CC, and (**l**) SC1200/CC anode biofilms at MFCs after 54 d of batch mode operations.

**Figure 9 membranes-12-00120-f009:**
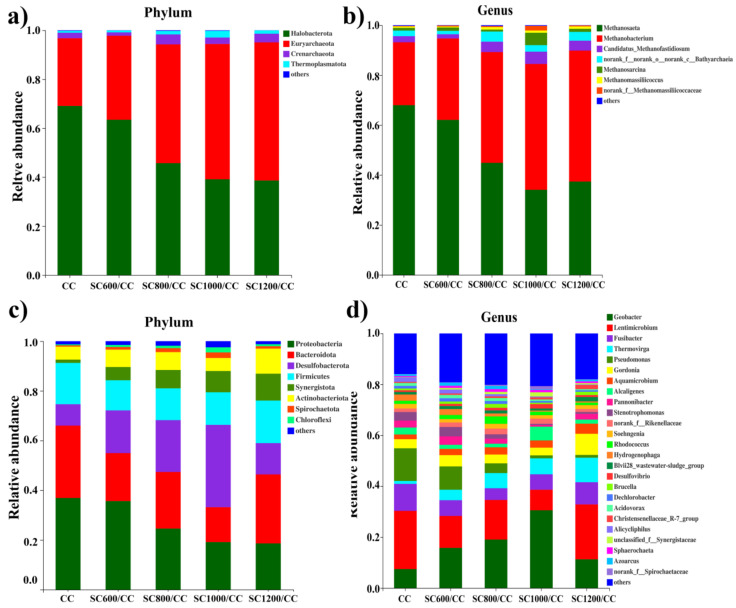
Microbial community structures of anodic biofilms of Archaea and bacteria attached on CC and SC anodes at the (**a**,**c**) phylum and (**b**,**d**) genus levels.

**Table 1 membranes-12-00120-t001:** Physical parameters of SC samples.

Samples	Yield (%)	pH	Zeta Potential (mV)	Ash (%)	Volatiles (%)	Conductivity (s/m)
SC600	66.33	8.09	−19.20	78.60	20.30	13.87
SC800	63.63	8.56	−10.59	81.90	15.50	36.40
SC1000	55.93	8.98	1.39	83.20	12.70	57.43
SC1200	57.27	9.65	2.14	86.70	9.80	54.77

## Data Availability

Not applicable.

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
