# Peer review of "Sludge Derived Carbon Modified Anode in Microbial Fuel Cell for Performance Improvement and Microbial Community Dynamics"

_membranes, 2022, doi:10.3390/membranes12020120_

Round 1
Reviewer 1 Report
The manuscript reported the effect of sludge carbon (SC) carbonized at temperatures of 600, 800, 1000, and 1200 °C on the anode performance of microbial fuel cells. The result demonstrates that the addition of SC improves microbial activity and optimizes microbial community structure, promotes the expression of c-type cytochromes, and is conducive to the formation of electroactive biofilm.
I consider the content of this manuscript will definitely meet the reading interests of the readers of the Membranes journal. However, there are certain English spelling and grammar issues, and also the discussion and explanation should be further improved.
Therefore, I suggest giving a minor revision and the authors need to clarify some issues or supply some more experimental data to enrich the content. This could be a comprehensive and meaningful work after revision.
My detailed comments can be found in a separate PDF file.

Reviewer 2 Report
The reviewer congratulates the authors on a quality manuscript with interesting and well-analysed results. Please, find below suggestions to be addressed by the authors:
Line 35-37 requires a reference. One for carbon materials in general and another one that justifies that carbon felt has small specific surface area.
Line 81 (Section 2.1). Please indicate the properties and characteristics of the carbon cloth used and the branch. Or indicate it in section 2.2. for anode and cathode.
Section 2.2. External resistance to close the electrical circuit is missing in the description under current operation conditions. Please, explain the inoculation procedure in detail.
Line 92: " All experiments were conducted in triplicates, and the average value was calculated." The authors comment that they have performed the experiments in a replicated manner, therefore, the reviewer recommends to put the data of the results with the corresponding deviations/errors.
Line 94-96. Please include a reference related to the procedure of carrying out polarization curves manually, because according to the description it can be understood that the procedure was manual.
Line 96. Missing the “T” in the sentence “he prepared anode…”. What equipment was used for the Cyclic voltammetry?
Line 120-121. Please add a reference.
Lines 136-138. Please add a reference.
Table 1 – pH = 8.09-9.65, is it not a too basic pH for electrogenic bacteria?
Line 205-206. Please add reference
Line 223-224. Please add reference
Figure 4. Size of the graphics makes interpretation difficult for readers
Figure 5. the reviewer strongly recommends using the same scale so that the difference between the different anodes can be appreciated.
Line 316 – How long takes a cycle? In this context, please indicate in Section 2.2 The MFC operation mode: continuous or discontinuous.
Section 3.3. Please, explain the trend of the polarization curves.
Line 544 is missing due to overlapping with figure 13.
Section 3.5. Reviewer recommends the authors to show the pH of the anodic liquor, in order to relate it with the bacteria detected in the microbial community analysis.
It would be interesting to carry out studies above 1200º of pyrolysis, to check that the higher the temperature, the worse it works.
Author Response
Please see yhe attachments

Round 2
Reviewer 2 Report
The reviewer thanks the authors for their efforts to improve the manuscript. However, additiona modifications are required:
- “smooth surface of graphite rod” requires a reference.
- “The substrates of anode and cathode are carbon cloth”. The term substrate can cause confusion, please modify it to material
- “Then connect the prepared electrode with an external resistance of 1000 Ω”. Please modify the sentence to be clearer. For example: Then, the prepared electrodes were connected by means of an external resistance of 1000 Ω.
- Please justify the use of 1000 Ω for the external resistance or add a reference.
- “use the data acquisition card to collect data every 10 minutes MFC adopts intermittent water inlet mode, that is, when the external voltage of the battery drops below 50 mV, replace it with new wastewater”. Please rewrite this sentence using a better English.
- Despite authors indicate that errors and deviations were included in the first version of the manuscript, the reviewer did not find the data expressed with errors and deviations or their inclusion in the graphs in either the first version or the present version. Please express the data as follows (as an example): 5.985±002 A·m-2 and include the deviations in the data graphics whenever possible. For example, figures 2A, 2B, 6A, 6B,6C, 6D,may include errors. This is critical.
